# Comprehensive Kinetic Study of PET Pyrolysis Using TGA

**DOI:** 10.3390/polym15143010

**Published:** 2023-07-11

**Authors:** Zaid Alhulaybi, Ibrahim Dubdub

**Affiliations:** Chemical Engineering Department, King Faisal University, P.O. Box 380, Al-Ahsa 31982, Saudi Arabia

**Keywords:** polyethylene terephthalate (PET), pyrolysis, thermogravimetric analysis (TGA), kinetics, thermodynamic parameters

## Abstract

The pyrolysis of polyethylene terephthalate (PET) is a well-known process for producing high fuel value. This paper aims to study the kinetics of PET pyrolysis reactions at 4 different heating rates (2, 5, 10, and 20 K min^−1^) using thermogravimetric analysis (TGA) data. TGA data show only one kinetic reaction within the temperature ranges of 650 to 750 K. Five different model-free models, namely, the Freidman (FR), Flynn–Wall–Qzawa (FWO), Kissinger–Akahira–Sunose (KAS), Starink (STK), and distributed activation energy model (DAEM), were fitted to the experimental data to obtain the activation energy (*E_a_*) and the pre-exponential factor (*A*_0_) of the reaction kinetics. The Coats–Redfern (CR) model equation was fitted with the help of master plot (Criado’s) to identify the most convenient reaction mechanism for this system. *E_a_*’s values were determined by the application of the five aforementioned models and were found to possess an average value of 212 kJ mol^−1^. The mechanism of PET pyrolysis reaction was best described by first-order reaction kinetics; this was confirmed by the compensation. Further thermodynamic parameter analysis indicated that the reaction was endothermic in nature.

## 1. Introduction

Plastic waste does not biodegrade easily and its steady increase in the last three decades constitutes a major concern for its environmental risk. Recycling of this great amount of waste has been investigated by many countries, scientists, and researchers, and pyrolysis degradation is one of the most promising solutions for the recovery of a high-value fuel product. TGA analysis was usually run at the first stage to collect the various kinetic parameters which are essential for reaction process design.

PET is recognized as one of the main six polymers in plastic waste, and about 7.6% of plastic waste is PET (Martín-Gullón et al. (2001) [1], Diaz-Silvarrey et al. (2018) [2]). PET is used in the manufacturing of fibers and films, especially for soft drink containers, because of its high stability characteristics (Moltó et al. (2007) [3], Çepelioğullar and Pütün (2013) [4]). Many researchers have reported the results of kinetics studies on PET pyrolysis reaction using TGA in the last twenty years. Some applied a simple “curve-fitting“ method to collect the kinetic parameters and hence reported *E_a_* of 242 kJ mol^−1^ with one reaction order (Yang et al. (2001) [5]). Subsequently, the decomposition kinetics of PET polymer were studied by the isoconversional method and the Vyazovkin model-free method, and the activation energy was found to be strongly dependent on the conversion of reaction (Saha et al. (2006) [6]). Brems et al. (2011) [7] reported the average value of activation energy of 237 kJ mol^−1^ at wide ranges of heating rate (3–120 K min^−1^) for PET pyrolysis reaction, similar to the value obtained by FWO model-fitting calculation.

Çepelioğullar and Pütün (2013) [4] studied the pyrolysis of PET alone as a set of experiments. They used CR (model fitting) at a single heating rate of 10 K min^−1^ to calculate the kinetic parameters. They found 2 values of activation energy (347.4 kJ mol^−1^ for conversion of 0.4–81% and 172.6 kJ mol^−1^ for the conversion of 88–99%, respectively). They attributed this high value of activation energy to the complex compound with an aromatic ring structure. Diaz Silvarrey and Phan (2016) [8] carried out TGA analysis for the pyrolysis of PET at different heating rates (5, 10, 20, and 40 K min^−1^). They determined the value of *E_a_* and the pre-exponential factor of 197.61 kJ mol^−1^ and 4.84 × 1014 s^−1^, respectively, using KAS and FR methods. Miandad et al. (2017) [9] found that the decomposition of PET occurred within 2 stages (peak 753 K and 923), results which contradicted the findings of Dimitrov et al. (2013) [10], and found that these differences were due to polymeric structure and the differences in degradation mechanisms (Chandrasekaran et al. (2015) [11]). Ganeshan et al. (2018) [12] applied the CR model to PET pyrolysis reaction at 3 different heating rates (15, 20, and 25 K min^−1^) with the 2-stage degradation profiles and found the *E_a_* values to be between 133 and 251 kJ mol^−1^. They highlighted that the main initial decomposition in the range of temperature (600–740 K) occurred with more than 80% mass loss. They pointed out that the CR model analysis was not appropriate for evaluating the kinetic parameters (Osman et al. (2020) [13]). Das and Tiwari (2019) [14] used plastic Coca-Cola bottle waste as a PET sample for TGA pyrolysis at a wide range of heating rates (5, 10, 20, 40, and 50 K/min). They used only the advanced isoconversional (AIC) method and obtained activation energy values between 203 and 355 kJ mol^−1^. Further, they applied Criado’s master plot technique to establish F1 as a possible mechanism model for inert TGA data. Osman et al. (2020) [13] applied differential FWO, integral FR isoconversional, and kinetic modelling (ASTM-E698) to calculate the kinetic triplet for PET pyrolysis. They found different activation energy values with the above three methods with an error rate of less than 15% between the lowest and the highest value. Mishra et al. (2019) [15] used different isoconversional model-free methods (FR, KAS, FWO, STK, and CR) to obtain the kinetic parameters for PET pyrolysis. Average activation energy values for all these methods are presented in results and discussion section. They applied the CR method for a single heating rate in order to estimate the order of the reaction by adopting various models including the Avrami and diffusion model. They also calculated enthalpy, entropy, Gibbs free energy, and frequency factor as part of thermodynamic analysis using activation energy values obtained from the STK method. Chowdhury et al. (2023) [16] studied PET pyrolysis using FR (model-free), CR, and Arrhenius methods (model fitting) at 3 different heating rates (10, 20, and 30 K min^−1^). They calculated very different ranges of activation energy values (FR = 3.31 to 8.79 kJ mol^−1^, CR = 1.05 × 104 kJ mol^−1^, and Arrhenius = 1278.88 kJ mol^−1^). Moreover, they computed the thermodynamic parameters using the Arrhenius and CR models.

This current study aimed to obtain full information on PET pyrolysis using TGA experimental data. The kinetic parameters and the mechanism of the pyrolysis were collected comprehensively by five isoconversional methods (FR, FWO, KAS, STK, and DAEM) with two nonisoconversional methods (CR and Criado). In addition, thermodynamic parameters of the pyrolysis reaction process were calculated and interpreted from the model analysis.

## 2. Materials and Procedures

### 2.1. The Proximate and Ultimate Analyses

PET was collected from the same supplier (Recycled Plastic, Ipoh, Malaysia) as used during our previous experiment (Dubdub and Al-Yaari (2020) [17]). Two main characteristic (proximate and ultimate) analyses were performed to obtain the physiochemical properties of the sample, for which detailed results are presented in Table 1. The procedure is detailed elsewhere (Dubdub and Al-Yaari (2020) [17]).

### 2.2. TGA of PET

PET polymer samples which were collected from Recycled Plastic (Ipoh, Malaysia), were ground into powder before use in the TGA analysis. An amount of 10 mg of PET was used with 40 mL min^−1^ N_2_ gas flow inert atmosphere at 4 different heating rates (2, 5, 10, and 20 K min^−1^). Each test run of these four is indicated by PETx throughout the paper, and x number refers to the heating rate. Multiple heating rates were applied, following the recommendations of the ICTAC (Koga et al. (2023) [19]). Mishra et al. (2019) [15] noted that for accurate determination of kinetic parameters, a low heating rate (below 8 K min^−1^) should be used, and the ratio between the highest and lowest heating rate should be greater than 10 K min^−1^ (Osman et al. (2020) [13]).

### 2.3. Derivation of the Kinetic Equations

The derivation of PET pyrolysis reaction will be based on the following well-known Arrhenius equation:(1)$$\frac{d\alpha }{dt}=A_{0}\exp\left(-\frac{E_{a}}{RT}\right)f\left(\alpha \right)$$
where *α* is the reaction conversion, *t* is time, *E_a_* is the activation energy, *A*_0_ is the frequency factor, *R* is the universal gas constant, and *T* is the absolute temperature (Dubdub and Al-Yaari (2020) [17]).

For nonisothermal test, *β* (heating rate) can be introduced in the above equation as:(2)$$\beta \frac{d\alpha }{dT}=A_{0}\exp\left(-\frac{E_{a}}{RT}\right)f\left(\alpha \right)$$

All five model-free methods can be derived from Equation (2) either integrally to obtain FR method or differentially to obtain FWO, KAS, STK, and DAEM with some assumptions for each method (Chowdhury et al. (2023) [16], Aboulkas et al. (2010) [20], Dubdub (2023) [21]). Table 2 presents these five isoconversional equations, and Table 3 shows the equations for CR and Criado. Criado attempted to verify the reduced theoretical curve (left side) and the experimental data (right side) in Equation (9). Therefore, a comparison between them will help us determine which kinetic model will describe the experimental reaction. Table 4 shows the common solid-state thermal reaction mechanisms *f*(*α*) and *g*(*α*) used in the CR and Criado method (Table 3).

By plotting ($\ln\left(\beta \frac{d\alpha }{dT}\right),\ln\left(\beta \right),\ln\left(\beta /T^{2}\right),\;\ln\left(\beta /T^{1.92}\right),\ln\left(\beta /T^{2}\right))$ against $\frac{1}{T}$ for FR, FWO, KAS, STK, and DAEM models, the value of *E_a_* will be obtained from the slope of the line. The obtained values of *E_a_* by these methods are independent of the reaction mechanism. CR model, expressed by Equation (9), was implemented to find the most convenient reaction mechanism from 15 options (Table 4). After that, the values of the *A*_0_ can be obtained from the slope of the linear relationships of Equations (3)–(7) when the reaction mechanism has been specified.

### 2.4. Thermodynamic Parameter Analysis of PET Pyrolysis

The various thermodynamic model parameters for the PET pyrolysis reaction based on the model fittings’ calculated values of (*E_a_*, *A*_0_, and *T_p_*), can be determined by the following three equations:(10)$$∆H=E_{a}-RT_{p}$$
(11)$$∆G=E_{a}+RT_{p}ln\left(\frac{k_{B}T_{p}}{hA}\right)$$
(12)$$∆S=\frac{∆H-∆G}{T_{p}}$$
where: ∆*H* is the change in enthalpy, ∆*G* is the change in Gibbs free energy, ∆*S* is the change in entropy, *T_p_* is the maximum peak temperature obtained from the derivative thermogravimetric curves, *k_B_* is the Boltzmann constant (1.381 × 10^−23^ J/K), and *h* is the Planck constant (6.626 × 10^−34^ J/s).

The thermodynamic parameters (∆H, ∆G, and ∆S) are of great importance to the optimization of the large-scale reactor used for pyrolysis. In addition, it is important to verify the energy and the suitability of PET pyrolysis process using the thermodynamic parameters (Dhyani et al. (2017) [22]).

## 3. Results and Discussion

### 3.1. TGA of PET

The thermogravimetric (TG) and derivative thermogravimetric (DTG) curves of the PET pyrolysis at 2, 5, 10, and 20 K min^−1^ heating rates are shown in Figure 1a,b. All curves show similarities in their trend, with shifting to the right (higher temperatures) as the heating rate increases and the mass loss at constant temperature decreases (Figure 1a). A higher heating rate means more energy will be added to the sample, pushing the process to occur at a higher rate and temperature. In addition, an increased heating rate may alter the kinetics of the PET pyrolysis process, which can change the characteristic temperatures (Chowdhury et al. (2023) [16]). Additionally, with the increasing heating rate, the DTG peak and mass loss rate also increased (Figure 1b). This behavior may be due to the heat transfer limitation or thermal lag (Al-Salem et al. (2017) [23]).

As shown in Figure 1, the pyrolysis of PET occurs in the temperature ranges of 650 to 750 K with about 20 wt% residue production. These two curves showed only one reaction stage, as reported elsewhere (Table 5). The characteristic temperatures of pyrolysis are presented in Table 5. This indicates that the characteristic temperatures (onset, peak, and final) found in this study were in agreement with many previous researchers’ reported work (Table 5). Table 5 also shows production of different residue amounts during PET pyrolysis, as reported by many researchers. PET pyrolysis represents the cross-linking of the products to produce more polyaromatics, responsible for char formation (Singh et al. (2020) [24]). Ganeshan et al. (2018) [12] observed 2 stages of reaction, with the first main initial decomposition occurring within 600–740 K with more than 80% mass loss.

### 3.2. Determination of Kinetic Parameters by Model-Free Methods

Five model-free models (FR, FWO, KAS, STK, and DAEM) were used here to calculate the value of *E_a_* from the TGA data set and from the slope of the fitting plots as shown in Table 1. Regression of all plots is also shown in Figure 2, while the values of *E_a_* and the obtained *R*^2^ value by the five models are displayed in Figure 3 and tabulated in Table 6 at a conversion range of 0.1–0.8. A low correlation (R2) value at a reaction conversion of 0.9 indicates inapplicability of this fitted model (Mishra et al. (2019) [15], Damartzis et al. (2011) [25]). The average activation energies obtained from FR, FWO, KAS, STK, and DARM methods were 204, 220, 211, 215, and 214 kJ mol^−1^, respectively. The correlation coefficient (*R*^2^) was found to be higher than 0.8 for most of the methods except at 2 conversion reactions (0.7 and 0.8) for the FR method, where the values of *R*^2^ were between 0.7 and 0.8. This low correlation coefficient value may be due to the end of the reaction.

The obtained *E_a_* values from all curve-fitting methods except the FR method were in very close agreement, indicating the superior applicability of all four other models compared to the FR method (Mishra et al. (2019) [15]). The final average activation value of 212 kJ mol^−1^ for the five methods was very close to the values obtained by many other researchers (Table 7). For example, Osman et al. (2020) [13] reported *E_a_* values of 165–195 kJ mol^−1^ by differential isoconversional methods and 166–180 kJ mol^−1^ by FWO. Chowdhury et al. (2023) [16] noted a very low value of *E_a_* (3.31–8.79 kJ mol^−1^) using the FR model, albeit with reasonably high values of *R*^2^ (0.8648–0.9567). Das and Tiwari (2019) [14] calculated *E_a_* by advance isoconversional method (AIC) at higher heating ranges (5–50 K min^−1^), and its value was found to be within the range of 203–355 kJ mol^−1^, with large variation over the conversion ranges of 0.05–0.8. They observed that the change of *E_a_* tended to start at a low value and increase over the duration of the pyrolysis reaction. Mishra et al. (2019) [15] obtained variation values of *E_a_* (KAS = 210–241 kJ mol^−1^, FWO = 211–241 kJ mol^−1^, STK = 211–242 kJ mol^−1^, and FR = 208.6–236.0 kJ mol^−1^) over reaction conversion of 0.1–0.8.

### 3.3. Determining the Kinetic Parameters by Model-Fitting Methods

The CR model, Equation (8), was applied to identify the suitable reaction mechanism/s for PET pyrolysis. Therefore, values of *E_a_* and *A*_0_ at different heating rates for 15 solid-state reaction mechanisms were determined from the linear fitting plots between ln(g(*α*)/*T*^2^) versus 1/*T*; the obtained kinetic parameters are presented in Table 8. As shown in Table 8, the CR method was well fitted for the TGA data of the PET pyrolysis, with a higher linear regression coefficient of *R*^2^ > 0.99.

Only one reaction demonstrated a straight line, and the values of *E_a_* calculated by CR with a function of g(*α*) (F1–P4) are shown in Table 8. A large deviation in the value of *E_a_* was found (in the range of 45–503 kJ min^−1^) in the case of different reaction mechanisms (F1–P4) that did not suit the PET pyrolysis reaction.

The correct value for these ranges of activations will be selected from the comparison between the right and left side of the Criado (shown in Figure 4 and Table 9) method. Most of the values of *E_a_*, except F1, were not considered because of their large deviation from the expected *E_a_* values. All these graphs, except F1, were deleted from Figure 4a,c,e,g because they were not close agreement with the experimental range values. Figure 4b,d,f,h show only the most controlling model reaction mechanism (reaction order models—first order—F1) (Das and Tiwari (2019) [13]). Table 9 lists *E_a_*, ln (*A*_0_), and *R*^2^ for all four tests. The “g(*α*)-F1”, as the final reaction mechanism, will be used with each of five model-free isoconversional methods (FR, FWO, KAS, STK, and DAEM) to obtain the value of ln(*A*_0_) (Table 10).

Some studies in the literature reported the direct use of the CR method alone, based on using one mechanism step prior to the model reaction. However, in this current research, the appropriate mechanistic model as a second trend was determined by the Criado model’s fitting approach. The CR model’s approach to a single heating rate could cause a high possibility of data failure (Chowdhury et al. (2023) [16]). Chowdhury et al. (2023) [16] used two nonisoconversional models (CR and Arrhenius) and found an increase in *E_a_* values with an increasing heating rate; this enhancement may be attributed to the increase in the reaction rate. They found abnormal values of *E_a_* (CR = 1.02 × 10^4^–1.05 × 10^4^ kJ mol^−1^, Arrhenius = 888.75–1889.94 kJ mol^−1^). Ganeshan et al. (2018) [12] applied the CR method alone to obtain *E_a_* and *A*_0_, assuming reaction orders between 0.1 and 3 and selecting the best *R*^2^ values at each heating rate. They observed that *E_a_* decreased with the increased in heating rate, which was not observed in our study. They found that the best value of the reaction order was 1.5 with *E_a_* = 244.496 kJ min^−1^ and *R*^2^ = 0.997 for PET pyrolysis.

Das and Tiwari (2019) [14] also determined the reaction mechanism by selecting the best linear fitting curves for each heating rate between the theoretical and the experimental value. They detected that all the models with equal linearity coefficient *R*^2^ were represented as A2, A3, A4, and F1. Then, among the four models, F1 was identified as the best model, producing the lowest root-mean-square error (RMSE).

The selected mechanism by the Criado method or compensation effect (Equation (9)) could be used to determine the linearity between *lnA*_0_ and *E_a_*. Figure 5 shows the linear relationship (*R*^2^ = 0.9991). This improves the convenience of the suggested model for PET pyrolysis. Das and Tiwari (2019) [14] confirmed that the values *A*_0_ can compensate for the values of *E_a_* by obtaining a straight line from a plot of $lnA_{0}$ with *E_a_* over the entire conversion range.

### 3.4. Thermodynamic Parameter Analysis

The thermodynamic parameters (∆H, ∆G, and ∆S) were calculated for (for 2, 5, 10, and 20 K min^−1^) and are shown Table 11.

Table 11 shows positive values of ∆H (206.34, 206.3, 206.1, and 206.0 kJ mol^−1^ at 2, 5, 10, and 20 K min^−1^, respectively), indicating that the main reaction was endothermic in nature. A similar observation was reported by Chowdhury et al. (2023) [16].

The positive and negatives values of ∆G and ∆S indicate that the PET pyrolysis was a nonspontaneous process in nature. The negative signs of ∆*S* at different heating rates indicate that the process became less disturbing in thermodynamic equilibrium and low in reactivity (Chowdhury et al. (2023) [16]). The thermodynamic results reveal the promising potential of PET pyrolysis to be efficiently used to produce bioenergy. Chowdhury et al. (2023) [16] applied the kinetics parameters of the CR model to determine thermodynamic parameter values at different heating rates.

As long as pyrolysis is a process used to produce high-energy fuel, our comprehension of changes in enthalpy, entropy, and free energy with different heating rates must consider thermodynamics (Enyoh et al. (2022) [28], Xu and Chen (2013) [29]). Chowdhury et al. (2023) [16] computed the thermodynamic parameters using the kinetics parameters obtained by the CR and Arrhenius models.

## 4. Conclusions

Pyrolysis of PET at various heating rates of 2, 5, 10, and 20 K min^−1^ was investigated and the results are discussed here. TG and DTG curves indicated that PET pyrolysis occurred in 1 stage covering the temperature range of 650–750 K. TGA experimental data were fitted well by the five most-popular (differential and integral) model-free isoconversional models (FR, FWO, KAS, STK, and DAEM), and the values of *E_a_* and the pre-exponential factor for PET pyrolysis were determined. However, the most appropriate reaction mechanism was fixed by the CR nonisoconversional model and the master plot of the Criado method. The appropriateness of the mechanisms was shown by the linearity of the relation between *lnA*_0_ and *E_a_*.

The thermodynamic properties for PET pyrolysis reaction at different heating rates showed that this reaction was endothermic in nature and confirmed the suitability for the production of bioenergy by the pyrolysis process.

## Figures and Tables

**Figure 1 polymers-15-03010-f001:**
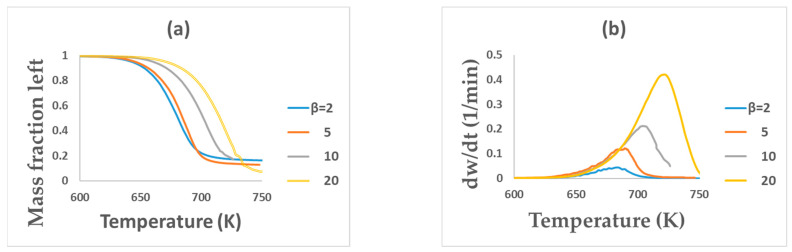
TG (**a**) and DTG (**b**) curves of PET pyrolysis at different heating rates.

**Figure 2 polymers-15-03010-f002:**
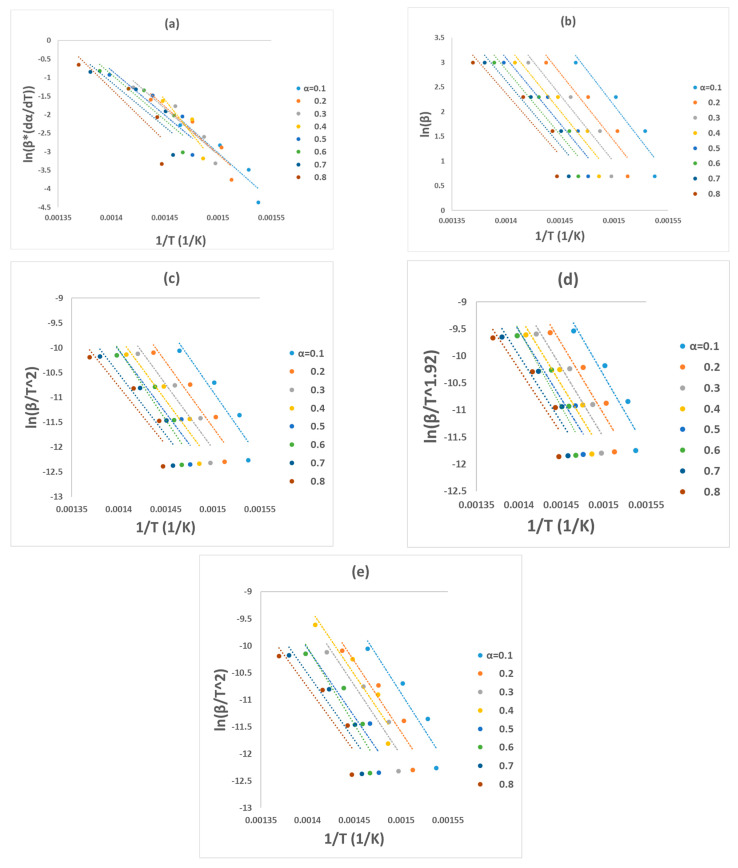
Regression lines of the experimental data of PET pyrolysis by (**a**) FR, (**b**) FWO, (**c**) KAS (**d**) STK, and (**e**) DAEM models.

**Figure 3 polymers-15-03010-f003:**
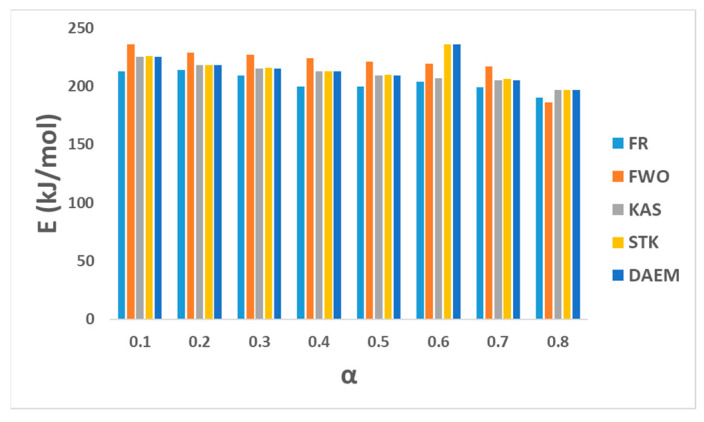
Regression lines of the experimental data of PET pyrolysis.

**Figure 4 polymers-15-03010-f004:**
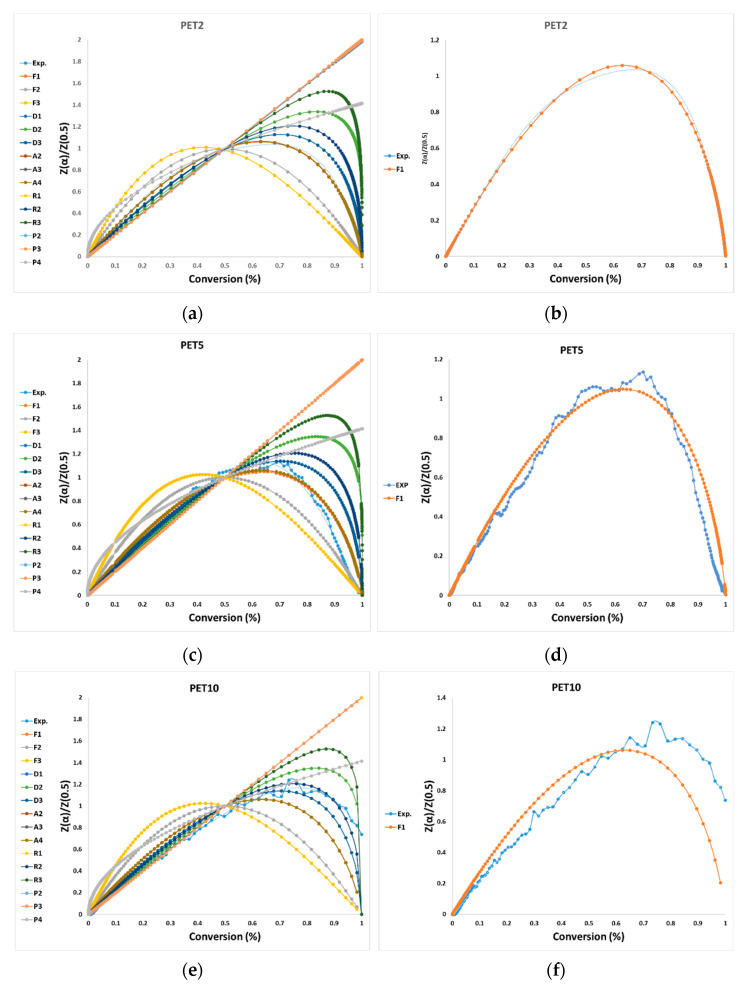
Master plots of different kinetic models and experimental data of four tests; (**a**,**b**)PET2, (**c**,**d**) PET5, (**e**,**f**) PET10, and (**g**,**h**) PET20.

**Figure 5 polymers-15-03010-f005:**
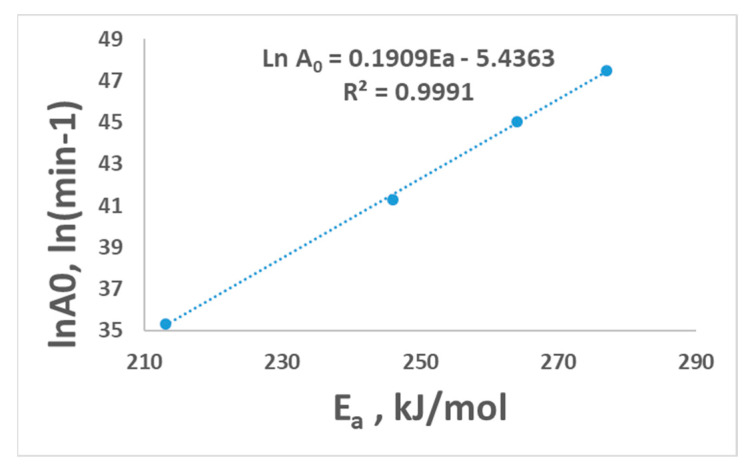
Linear fitted curve for the compensation effect.

**Table 1 polymers-15-03010-t001:** The proximate and ultimate analysis of PET (Dubdub and Alhulaybi (2023) [18]).

Proximate Analysis, wt%			Ultimate Analysis, wt%			
Moisture	Volatile	Ash	C	H	N	O
0.523	88.231	11.246	64.256	4.367	0	31.377

**Table 2 polymers-15-03010-t002:** Equations for model-free methods (Dubdub (2023) [20]).

Method	Equation		Integral (I) or Differential (D)	Plot
FR	$\ln\left(\beta \frac{d\alpha }{dT}\right)=\ln\left[A_{0}f(\alpha \right)]-\frac{E_{a}}{RT}$	(3)	D	$\ln\left(\beta \frac{d\alpha }{dT}\right)\;vs.\;\frac{1}{T}$
FWO	$\ln\left(\beta \right)=ln\frac{A_{0}E_{a}}{Rg(\alpha )}-5.331-1.052\frac{E_{a}}{RT}$	(4)	I	$\ln\left(\beta \right)\;vs.\;\frac{1}{T}$
KAS	$\ln\left(\frac{\beta }{T^{2}}\right)=ln\frac{A_{0}R}{E\;g(\alpha )}-\frac{E_{a}}{RT}$	(5)	I	$\ln\left(\beta /T^{2}\right)\;vs.\;\;\frac{1}{T}$
STK	$ln\frac{\beta }{T^{1.92}}=\;ln\left(\frac{A_{0}E_{a}}{Rg(\alpha )}\right)-1.0008\frac{E_{a}}{RT}$	(6)	I	$\ln\left(\beta /T^{1.92}\right)\;vs.\;\;\frac{1}{T}$
DAEM	$ln\frac{\beta }{T^{2}}=\;ln\left(\frac{A_{0}R}{E_{a}}\right)+0.6075-\frac{E_{a}}{RT}$	(7)	I	$\ln\left(\beta /T^{2}\right)\;vs.\;\;\frac{1}{T}$

**Table 3 polymers-15-03010-t003:** Equations for model-fitting methods (Dubdub (2023) [21]).

Method	Equation	
**CR**	$ln\left[\frac{g(\alpha )}{T^{2}}\right]=ln\left[\frac{A_{0}R}{\beta E}\right]-\frac{E}{RT}$	(8)
**Criado**	$\frac{Z(\alpha )}{Z(0.5)}=\frac{f\left(\alpha \right)g(\alpha )}{f\left(0.5\right)g(0.5)}={\left(\frac{T_{\alpha }}{T_{0.5}}\right)}^{2}\frac{{\left(\frac{d\alpha }{dt}\right)}_{\alpha }}{{\left(\frac{d\alpha }{dt}\right)}_{0.5}}$	(9)

**Table 4 polymers-15-03010-t004:** Solid-state thermal reaction mechanism (Dubdub (2023) [21]).

Reaction Mechanism	Code	f(α)	g(α)
Reaction order models—first order	F1	1 − *α*	−ln⁡(1−α)
Reaction order models—second order	F2	${\left(1-\alpha \right)}^{2}$	${\left(1-\alpha \right)}^{-1}-1$
Reaction order models—third order	F3	${\left(1-\alpha \right)}^{3}$	${\left[(1-\alpha \right)}^{-1}-1]/2$
Diffusion model—one dimension	D1	$1/2\alpha^{-1}$	$\alpha^{2}$
Diffusion model—two dimensions	D2	${\left[-ln(1-\alpha )\right]}^{-1}$	$\left(1-\alpha \right)\ln\left(1-\alpha \right)+\alpha$
Diffusion model—three dimensions	D3	$3/2{\left[1-{\left(1-\alpha \right)}^{1/3}\right]}^{-1}$	${\left[1-{\left(1-\alpha \right)}^{1/3}\right]}^{2}$
Nucleation models—two dimensions	A2	$2(1-\alpha ){\left[-\ln\left(1-\alpha \right)\right]}^{1/2}$	${\left[-\ln\left(1-\alpha \right)\right]}^{1/2}$
Nucleation models—three dimensions	A3	$3(1-\alpha ){\left[-\ln\left(1-\alpha \right)\right]}^{1/3}$	${\left[-\ln\left(1-\alpha \right)\right]}^{1/3}$
Nucleation models—four dimensions	A4	$4(1-\alpha ){\left[-\ln\left(1-\alpha \right)\right]}^{1/4}$	${\left[-\ln\left(1-\alpha \right)\right]}^{1/4}$
Geometrical contraction models—one dimension	R1	1	α
Geometrical contraction models—sphere	R2	$2{\left(1-\alpha \right)}^{1/2}$	$1-{\left(1-\alpha \right)}^{1/2}$
Geometrical contraction models—cylinder	R3	$3{\left(1-\alpha \right)}^{1/3}$	$1-{\left(1-\alpha \right)}^{1/3}$
Nucleation models—two-power law	P2	$2\alpha^{1/2}$	$\alpha^{1/2}$
Nucleation models—three-power law	P3	$3\alpha^{2/3}$	$\alpha^{1/3}$
Nucleation models—four-power law	P4	$4\alpha^{3/4}$	$\alpha^{1/4}$

**Table 5 polymers-15-03010-t005:** Comparison of pyrolysis characteristics of PET at different heating rates between current work and earlier reported studies by various investigators.

Heating Rate K/min	This Work				Previous Work				References
	Onset (K)	Peak (K)	Final (K)	Mass Loss (%)	Onset (K)	Peak (K)	Final (K)	Mass Loss (%)	
2	620	680	690	80	623	667	694	100	Osman et al. (2020) [13]
5	650	685	710	85	658 623	700 673	723 733	80 80	Das and Tiwari (2019) [14] Diaz Silvarrey and Phan (2016) [8]
10	670	710	725	80	671 643 633 643 585	711 714 700 693 648 707	748 775 773 743 856	80 80 79.78 80 27.5	Das and Tiwari (2019) [14] Yang et al. (2001) [5] Çepelioğullar and Pütün (2013) [4] Diaz Silvarrey and Phan (2016) [8] Mishra et al. (2019) [15] Chowdhury et al. (2023) [16]
20	680	725	750	95	681 673 603	721 703 713	759 773 882	80 80 22.5	Das and Tiwari (2019) [14] Diaz Silvarrey and Phan (2016) [8] Chowdhury et al. (2023) [16]
30					661	672 716	878	27.5	Mishra et al. (2019) [15] Chowdhury et al. (2023) [16]
40					690 698 693	733 748 723	783 798 803	80 82 80	Das and Tiwari (2019) [14] Singh et al. (2020) [24] Diaz Silvarrey and Phan (2016) [8]
50					698	743 684	793	80	Das and Tiwari (2019) [14] Mishra et al. (2019) [15]

**Table 6 polymers-15-03010-t006:** Activation energy values obtained by five model-free methods.

Conversion	FR		FWO		KAS		STK		DAEM		Average	
	*E* kJ mol^−1^	*R* ^2^	*E* kJ mol^−1^	*R* ^2^	*E* kJ mol^−1^	*R* ^2^	*E* kJ mol^−1^	*R* ^2^	*E* kJ mol^−1^	*R* ^2^	*E* kJ mol^−1^	*R* ^2^
0.1	213	0.8777	236	0.9005	225	0.8914	226	0.8918	225	0.8914	225	0.8899
0.2	214	0.8858	229	0.898	218	0.8882	218	0.8886	218	0.8882	219	0.8912
0.3	209	0.9033	227	0.9047	215	0.8952	216	0.8957	215	0.8952	216	0.8986
0.4	200	0.8421	224	0.9037	213	0.894	213	0.8944	213	0.894	213	0.8838
0.5	200	0.8342	221	0.8953	209	0.8846	210	0.895	209	0.8846	210	0.8727
0.6	204	0.8314	219	0.8848	207	0.8546	236	0.8551	236	0.8546	220	0.8571
0.7	199	0.7583	217	0.8725	205	0.8594	206	0.8599	205	0.8594	206	0.8340
0.8	190	0.7646	186	0.8366	197	0.8198	197	0.8206	197	0.8198	193	0.8104
**Average**	**204**	**0.8372**	**220**	**0.8870**	**211**	**0.8734**	**215**	**0.8751**	**214**	**0.8734**	**212**	**0.8692**

**Table 7 polymers-15-03010-t007:** Activation energies from different published papers.

References	*E* (kJ mol^−1^)	Method
Yang et al. (2001) [5]	242	Curve fitting
Senneca et al. (2004) [26]	217	FR
Girij et al. (2005) [27]	227 208 236	FR Ozawa Kissinger
Saha et al. (2006) [6]	180–208	VY
Das and Tiwari (2019) [14]	203–355	AIC
Osman et al. (2020) [13]	165.6 166–180 165–195	ASTME698 FWO FR
Mishra et al. (2019) [15]	225.64 230.71 230.55 231.03	FR KAS FWO STK

**Table 8 polymers-15-03010-t008:** Kinetic parameters obtained by CR model.

Reaction Mechanism	Code	PET2			PET5		
		*E_a_* kJ mol^−1^	*ln*(*A*_0_)	R^2^	*E_a_* kJ mol^−1^	*ln*(*A*_0_)	R^2^
Reaction order models—first order	F1	246	41.3	0.9996	277	47.51	0.9999
Reaction order models—second order	F2	286	48.81	0.9989	393	68.73	0.9988
Reaction order models—third order	F3	330	57.01	0.998	533	94.14	0.9966
Diffusion models—one dimension	D1	432	73.87	0.9997	384	65.65	0.9983
Diffusion models—two dimensions	D2	454	77.43	0.9997	437	74.58	0.9992
Diffusion models—three dimensions	D3	479	80.49	0.9997	500	84.55	0.9998
Diffusion models—four dimensions	D4	465	77.45	0.9997	458	73.66	0.9995
Nucleation models—two dimensions	A2	118	17.91	0.9995	133	21.51	0.9999
Nucleation models—three dimensions	A3	75	9.82	0.9995	85	12.64	0.9999
Nucleation models—four dimensions	A4	53	13.12	0.9994	61	12.93	0.9999
Geometrical contraction models—one-dimension phase boundary	R1	210	34.49	0.9997	186	30.75	0.9982
Geometrical contraction models—sphere	R2	228	39.11	0.9997	229	37.89	0.9995
Geometrical contraction models—cylinder	R3	234	37.86	0.9997	244	40.34	0.9997
Nucleation models—two-power law	P2	100	14.41	0.9996	88	12.91	0.998
Nucleation models—three-power law	P3	63	11.74	0.9996	55	14.06	0.9977
Nucleation models—four-power law	P4	44	14.6	0.9995	38	16.54	0.9973
**Reaction Mechanism**	**Code**	**PET10**			**PET20**		
		** *E_a_* ** **kJ mol^−1^**	** *ln* ** **(*A*_0_)**	** *R* ** ** ^2^ **	** *E_a_* ** **kJ mol^−1^**	** *ln* ** **(*A*_0_)**	** *R* ** ** ^2^ **
Reaction order models—first order	F1	264	45.05	0.9996	213	35.33	0.9999
Reaction order models—second order	F2	373	64.5	0.9976	228	38.17	0.9996
Reaction order models—third order	F3	503	87.72	0.9949	244	41.13	0.999
Diffusion models—one dimension	D1	371	62.68	9996	408	67.96	1
Diffusion models—two dimensions	D2	420	70.9	0.9999	417	69	1
Diffusion models—three dimensions	D3	479	79.97	0.9999	427	69.3	1
Diffusion models—four dimensions	D4	440	72.9	1	421	68.1	1
Nucleation models—two dimensions	A2	127	20.58	0.9996	101	15.96	0.9999
Nucleation models—three dimensions	A3	81	12.22	0.9996	63	14.57	0.9999
Nucleation models—four dimensions	A4	58	14.37	0.9995	45	17.34	0.9999
Geometrical contraction models—one-dimension phase boundary	R1	180	29.58	0.9996	198	32.62	1
Geometrical contraction models—sphere	R2	219	36.12	1	205	33.27	1
Geometrical contraction models—cylinder	R3	234	38.35	0.9999	208	33.32	1
Nucleation models—two-power law	P2	84	12.63	0.9995	93	14.56	1
Nucleation models—three-power law	P3	52	15.34	0.9994	58	15.37	1
Nucleation models—four-power law	P4	36	17.67	0.9993	41	17.91	1

**Table 9 polymers-15-03010-t009:** Activation energy of (CR and Criado).

Test No.	*E_a_* kJ mol^−1^	*ln*(*A*_0_)	*R* ^2^	Reaction Mechanism
PET2	246	41.3	0.9996	Reaction order models—first order—F1
PET5	277	47.51	0.9999	Reaction order models—first order—F1
PET10	264	45.05	0.9996	Reaction order models—first order—F1
PET20	213	35.33	0.9999	Reaction order models—first order—F1

**Table 10 polymers-15-03010-t010:** Pre-exponential factor values obtained by isoconversional models.

Conversion		*ln*[*A*_0_ (min^−1^)]				
	FR	FWO	KAS	STK	DAEM	Average
0.1	35.45	37.56	17.27	17.87	39.33	29.50
0.2	35.84	36.31	15.98	16.58	37.22	28.39
0.3	34.94	35.99	15.65	16.25	36.4	27.85
0.4	33.46	35.59	15.23	15.84	35.6	27.14
0.5	33.63	35.05	14.69	15.28	34.71	26.67
0.6	34.39	34.74	19.44	19.91	39.3	29.56
0.7	33.53	34.49	14.09	14.69	33.53	26.07
0.8	39.51	33.18	12.77	13.38	31.83	26.13
Average	35.09	35.36	15.64	16.23	35.99	27.66

**Table 11 polymers-15-03010-t011:** Thermodynamic parameters.

Heating Rates (K/min)	2	5	10	20
**Kinetic Parameters**				
*E_a_* (kJ/mol)	212			
*A* (min^−1^)	1.29 × 10^11^			
*T_p_* (K)	680	685	710	725
**Thermodynamic Parameters**				
∆*H* (kJ/mol)	206.34	206.3	206.1	206.00
∆*G* (kJ/mol)	215.4	215.48	215.82	216.03
∆*S* (kJ/mol.K)	−0.01332	−0.0134	−0.01369	−0.01383
**Potential Energy Barrier**				
*E_a_*–∆*H* (kJ/mol) *	5.82			

* Based on the mean values of ∆H.

## Data Availability

Unavailable due to privacy or ethical restrictions.

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
