# Peer review of "Comprehensive Kinetic Study of PET Pyrolysis Using TGA"

_polymers, 2023, doi:10.3390/polym15143010_

Round 1

Reviewer 1 Report

Dear Editor,

The article aims to study the kinetics of PET pyrolysis experiments at four different heating rates (2, 5, 10, and 20 K min-1) using thermogravimetric analysis (TGA) data. The study requires extensive review before publication.

Introduction section is poorly written, the authors have only explained previous results in very detail, that is totally unnecessary. Please remove the text from line 43 to 78. Include novelty paragraph. Also explain what is is novel in this study. Rewrite this whole section or move it to the discussion section.

-Rename the heading 2.1 as Proximate and Ultimate analysis.

-rewrite line 95.

-delete line 191 to 193.

-Improve the English of the article. there are several serious mistakes in the text. 

-Improve the English of the article. there are several serious mistakes in the text. Revise the whole article for academic English writing.

Author Response

Responses to Reviewers and Editors comments

Comments and Suggestions for Authors

Dear Editor,

1-The article aims to study the kinetics of PET pyrolysis experiments at four different heating rates (2, 5, 10, and 20 K min-1) using thermogravimetric analysis (TGA) data. The study requires extensive review before publication.

Responses:      Thank you very much for this comment

Action:                       

All my published papers tried do proper "extensive review" by mentioned only the papers related to the title. For "Kinetic of PET by TGA", I tried to cover this title in the current paper, and if you have any more papers, I will be grateful and happy to add them.

2-Introduction section is poorly written, the authors have only explained previous results in very detail, that is totally unnecessary. Please remove the text from line 43 to 78. Include novelty paragraph. Also explain what is is novel in this study. Rewrite this whole section or move it to the discussion section.

Responses:      Thank you very much for this comment

Action:           

From my knowledge, correct me please if I am wrong, the contents of the introduction usually list all the papers related directly to the paper, and maybe will be used in the discussion in supporting some issues in the discussion. About the novelty, it is common to end up the introduction with this point:

"This current study was targeted mainly to have the full information on PET pyrolysis using TGA experimental data. The kinetic parameters and the mechanism of the pyrolysis were collected comprehensively by five isoconversional methods (FR, FWO, KAS, STK, and DAEM), with two non-isoconversional methods (CR and Criado). In addition, thermodynamic parameters of the pyrolysis reaction process have been calculated and interpreted from the model analysis."

3-Rename the heading 2.1 as Proximate and Ultimate analysis.

Responses:      Thank you very much for this comment

Action:           

Done according to your suggestion

4-rewrite line 95.

Responses:      Thank you very much for this comment

Action:           

Done according to your suggestion

5-delete line 191 to 193.

Responses:      Thank you very much for this comment

Action: 

-Improve the English of the article. there are several serious mistakes in the text. 

Comments on the Quality of English Language

-Improve the English of the article. there are several serious mistakes in the text. Revise the whole article for academic English writing.

Responses:      Thank you very much for this comment

Action:                        Surely, we will consider the quality English language. Since I have been advised to do English editing, the paper has been English/grammar checked thoroughly by an external English expert. 

Thanking you

Dr Ibrahim Dubdub

Reviewer 2 Report

The work is interesting and is of great importance in the study of the reaction kinetics of polyester pyrolysis. This is a work that could have been helpful to other scientists in the qualitative and quantitative assessment of the PET pyrolysis process if it had been described in more detail, especially in the procedure of the analyzes and calculations carried out. Therefore, in my opinion, the work should be improved or supplemented with the necessary information:

1. All abbreviations used in the text, including kinetic model abbreviations, should be explained in the text.
2. Why were these isoconversion and non-isoconversion methods chosen? Please provide a reason.
3. What software was used for calculations/simulations? Please specify in the text.
4. Chapter 2.1 on the two main analyzes is very sparse and cursory. The reader knows nothing about either the origin of the samples or these analyses. The whole point should be described in detail and then cited.
5. Section 3.1 presents the results of the main analysis. Nothing is known about the conditions of the analysis, and the apparatus used, and there is no comment on these results. How do these results relate to other results? simulated pyrolysis mechanisms?
6. The quality of drawing No. 1 is terrible. Please draw it in another graphics program or improve its quality.
7. Error bars are missing in Figure 3.
8. It was indicated that the pyrolysis process proceeded in 1 stage in the range of 650-750 K. The best kinetic model describing the course of pyrolysis was indicated. Therefore, a drawing should be made with the experimental TG curves and those determined by the mathematical method, illustratively indicating the adjustment made according to these two models to the experimental values. Was this matching according to the CR and Crado model performed over the entire measuring range or only at 650-750 K?

Author Response

Responses to Reviewers and Editors comments

Reviewer #2:

Comments and Suggestions for Authors

The work is interesting and is of great importance in the study of the reaction kinetics of polyester pyrolysis. This is a work that could have been helpful to other scientists in the qualitative and quantitative assessment of the PET pyrolysis process if it had been described in more detail, especially in the procedure of the analyzes and calculations carried out. Therefore, in my opinion, the work should be improved or supplemented with the necessary information:

  1. All abbreviations used in the text, including kinetic model abbreviations, should be explained in the text.

Responses:      Thank you very much for this comment

Action:

All abbreviations are explained.

  1. Why were these isoconversion and non-isoconversion methods chosen? Please provide a reason.

Responses:      Thank you very much for this comment

Action:

Isoconversional methods can be used only when there are more than one heating rate experiments (minimum three runs), because we will consider some parameters at constant conversion at different heating rate. While non-isoconversional methods can be used only for one heating rate, but unfortunately it is less accurate than the first.

Most of the published papers with different heating rates uses (could be considered as "Standard") mainly isoconversional methods in finding the main kinetic parameters (activation energy value Ea), and then move to non-methods to check the mechanism of the reactions (F1-P4) and calculate the rest of kinetic parameters (frequency factor value A0) depending on the proper selected reaction mechanism.

  1. What software was used for calculations/simulations? Please specify in the text.

Responses:      Thank you very much for this comment

Action: 

All the calculations have been done by Excel software (some tools inside the software). All the calculation could be sent by request.

  1. Chapter 2.1 on the two main analyzes is very sparse and cursory. The reader knows nothing about either the origin of the samples or these analyses. The whole point should be described in detail and then cited.

Responses:      Thank you very much for this comment

Action:

This section 2.1 has been rephrased and accordingly section 3.1 has been deleted. For more details for these two analyses method, please check the following paper:

https://doi.org/10.3390/polym15010070

  1. Section 3.1 presents the results of the main analysis. Nothing is known about the conditions of the analysis, and the apparatus used, and there is no comment on these results. How do these results relate to other results? simulated pyrolysis mechanisms?

Responses:      Thank you very much for this comment

Action:

As mentioned above section 3.1 has been deleted. For more details for these two analyses method, please check the following paper:

https://doi.org/10.3390/polym15010070

  1. The quality of drawing No. 1 is terrible. Please draw it in another graphics program or improve its quality.

Responses:      Thank you very much for this comment

Action:

Figure 1 has been redrawn

  1. Error bars are missing in Figure 3.

Responses:      Thank you very much for this comment

Action:

Usually, these graphs are presented in the comparison between different methods without error bar. I have done and attached below

  1. It was indicated that the pyrolysis process proceeded in 1 stage in the range of 650-750 K. The best kinetic model describing the course of pyrolysis was indicated. Therefore, a drawing should be made with the experimental TG curves and those determined by the mathematical method, illustratively indicating the adjustment made according to these two models to the experimental values. Was this matching according to the CR and Crado model performed over the entire measuring range or only at 650-750 K?

Responses:      Thank you very much for this comment

Action:

I agreed totally with your suggestion by creating new graphs showing the comparison between the experimental and the suggested models, but unfortunately, my trend in my published papers stopped at this stage, while some other (only few) researchers present this type of comparison.

Thanking you

Dr Ibrahim Dubdub

Round 2

Reviewer 2 Report

Accept in present form

Author Response

Thank you very much for your time to review the paper